# Simultaneous Determination and Health Risk Assessment of Four High Detection Rate Pesticide Residues in Pu’er Tea from Yunnan, China

**DOI:** 10.3390/molecules27031053

**Published:** 2022-02-04

**Authors:** Tao Lin, Xing-Lian Chen, Jin Guo, Meng-Xia Li, Yu-Feng Tang, Mao-Xuan Li, Yan-Gang Li, Long Cheng, Hong-Cheng Liu

**Affiliations:** 1Quality Standards and Testing Technology Research Institute, Yunnan Academy of Agricultural Science, Kunming 650223, China; lintaonj@126.com (T.L.); chen544141152@163.com (X.-L.C.); limx1970@126.com (M.-X.L.); yglikm@sina.com (Y.-G.L.); 2Laboratory of Quality and Safety Risk Assessment for Agro-Products (Kunming), Ministry of Agriculture and Rural Affairs, Kunming 650223, China; 3School of Medicine, Yunnan University of Business Management, Kunming 650106, China; gjin0111@163.com (J.G.); limengxia2022@163.com (M.-X.L.); 4College of Agronomy and Life Sciences, Zhaotong University, Zhaotong 657000, China; tyfsunny@163.com; 5SCIEX Analytical Instrument Trading Co., Ltd., Shanghai 200335, China; long.cheng@sciex.com

**Keywords:** high detection rate pesticides, multiwalled carbon nanotubes, risk assessment, Pu’er tea

## Abstract

Four pesticides with a high detection rate in Pu’er tea have been determined by a QuEChERS (quick, easy, cheap, effective, rugged, safe) method with multiwalled carbon nanotubes (MWCNTs), and combined ultrahigh-performance liquid chromatography–triple quadrupole linear ion trap-tandem mass spectrometry (UHPLC-QTRAP-MS/MS). MWCNs have been compared with other common purification materials, and found to be superior. The matrix effect was systematically studied, and the results show that the MWCNs can quickly and effectively reduce matrix interference values, which were in the range from −17.8 to 13.8. The coefficients (*R^2^*) were greater than 0.99, with the limit of quantification ranging from 0.1 to 0.5 μg/kg, and the recovery rate ranging from 74.8% to 105.0%, while the relative standard deviation (RSD) ranged from 3.9% to 6.6%. A total of 300 samples, taken from three areas in which Yunnan Pu’er tea was most commonly produced, tested for four pesticides. The results show that the detection rate of tolfenpyrad in Pu’er tea was 35.7%, which is higher than other pesticides, and the lowest was indoxacarb, with 5.2%. The residual concentrations of chlorpyrifos, triazophos, tolfenpyrad and indoxacarb ranged from 1.10 to 5.28, 0.014 to 0.103, 1.02 to 51.8, and 1.07 to 4.89 mg/kg, respectively. By comparing with China’s pesticide residue limits in tea (GB 2763-2021), the over standard rates of chlorpyrifos, tolfenpyrad, and indoxacarb were 4.35%, 0.87% and 0%, respectively. The risk assessment result obtained with the hazard quotient (HQ) method shows that the HQ of the four pesticides was far less than one, indicating that the risk is considered acceptable for the four pesticides in Pu’er tea. The largest HQ was found for tolfenpyrad, 0.0135, and the smallest was found for indoxacarb, 0.000757, but more attention should be paid to tolfenpyrad in daily diets in the future, because its detection rate, and residual and residual median were all relatively high.

## 1. Introduction

Pu’er tea is one of the most famous teas in China because of its various health benefits, such as lowering blood fat [1], weight loss [2], and antibacterial properties [3], as well as its unique taste and aroma [4,5]. It is beloved by consumers all over the world. With the increasing consumption of Pu’er tea [6], wild-grown tea raw materials can no longer meet the current sharp increase in consumption. To meet these demands, large-scale industrial farming of these ingredients, as well as the concurrent use of chemical pesticides, has been implemented. However, the indispensable and excessive use of pesticides has resulted in Pu’er tea containing many pesticide residues, which have potential risks to the health of its consumers [7].

Tolfenpyrad and indoxacarb are commonly used pesticides for Pu’er tea production or during tea plantation. They are effective in controlling *Empoasca pirisuga,* Matumura, and mite pests [8,9]. Indoxacarb is a low-toxicity pesticide [10], while tolfenpyrad is more toxic to aquatic organisms [11]. Chlorpyrifos and triazophos are also high detection rate pesticides. They are currently banned in China because they were detected in quantities exceeding the standard in vegetables, and also because of their moderate toxicity [12]. In addition, lower values of the acceptable daily intake (ADI) of chlorpyrifos and triazophos have been stipulated in the National Food Safety Standard—Maximum Residue Limits for Pesticides in Food (GB 2763-2021) [13].

In GB 2763-2021, the detection method for tolfenpyrad and indoxacarb in tea is liquid chromatography–tandem mass spectrometry, which requires the use of solid phase extraction cartridges for purification, and the use of toxic toluene as the elution solvent. This method is more harmful to the human body and the environment. The methods for the determination of chlorpyrifos and triazophos mainly include liquid chromatography–tandem mass spectrometry [14,15,16] and gas chromatography–mass spectrometry [17,18,19], but the detection period of gas chromatography–mass spectrometry is relatively long, which means that it cannot meet the current requirements for rapid high-sensitivity determination. When liquid chromatography–tandem mass spectrometry is used for the determination of low-level pesticides in complex matrices, false-positive detections may occur due to the interference of impurities [20,21]. The literature has reported the use of the QuEChERS method for the determination of pesticides in tea. Most of these methods, such as primary secondary amine (PSA), graphitized carbon black (GCB), and C_18_, are commonly used purification fillers [15,22,23,24]. However, there are few articles documenting the use of the QuEChERS method in Pu’er tea [25]. The same can be said for MWCNTs.

The complex matrix of Pu’er tea contains a variety of chemical components, which can affect the detection of pesticides. Conducting a qualitative analysis on the low levels of pesticides in Pu’er tea was difficult, especially when there was interference from the transitions or a shift in retention times [26,27]. Linear ion trap mass spectrometry combined a triple–quadruple (Qtrap) scanning functionality with sensitive ion trap scans. These additional Qtrap scan functions greatly enhanced the performance of screening, confirmation, and identification [20,21]. The detection rate of these four pesticide residues in Pu’er tea is high, indicating that human exposure to pesticides is becoming a more serious issue [28]. However, the current use of residue limits is not reasonable for the safety evaluation of Pu’er tea, and the lack of a multi-residue assessment method is unfavorable for the risk assessment of Pu’er tea.

In this study, the presence of four pesticides with a high detection rate in Pu’er tea was established using the QuEChERS method, combined with UHPLC/QTRAP-MS/MS and MWCNTs purifying filler. The established measurement method was used to determine the presence of pesticides in 300 samples from the three areas in which Yunnan Pu’er tea is most commonly produced. A risk assessment was carried out using the hazard quotient method [29]. The results provide basic data for consumers regarding the safety of drinking Pu’er tea.

## 2. Materials and Methods

### 2.1. Sample Collection and Preparation

A total of 300 Pu’er tea samples (500 g each) were collected in the Xishuangbanna, Pu’er and Lincang regions of Yunnan Province. A total of 100 samples from each region were collected. Sample collection was conducted in May 2021. All Pu’er tea samples were collected from supermarkets or farmers’ markets with the permission of local management personnel, and all the supermarkets or farmers’ markets are legally registered with their local authority. Collected samples were immediately transported to the laboratory and stored in the original sealed packaging at 4 °C in the dark. Sampling was carried out according to the Chinese standard (GB/T 8302-2013), established by the Standardization Administration of China [30].

### 2.2. Chemicals and Reagents

Chlorpyrifos, indoxacarb, triazophos, and tolfenpyrad standards were used at a concentration of 1000 mg/L, in which chlorpyrifos and triazophos were dissolved in acetone, and indoxacarb and tolfenpyrad were dissolved in methanol, purchased from Agro-Environmental Protection Institute, Ministry of Agriculture (Tianjin, China). Methanol and acetonitrile of HPLC grade were obtained from Merck KGaA (Darmstadt, Germany). Highly purified water was prepared by a Milli-Q water purification system (Bedford, MA, USA). Ammonium formate (≥ 99.995%) was purchased from Millipore Sigma Company (St Louis, MO, USA). Analytical reagent grade, including anhydrous sodium chloride (NaCl) and magnesium sulfate (MgSO_4_), were obtained from Sinopharm Chemical Reagent (Beijing, China). MWCNTs (10–20 nm diameter, 10–30 µm length) were provided by Nanjing XFNANO Materials Tech Co., Ltd. (Nanjing, China). Purification materials, including primary secondary amine (PSA), Florisil, graphitized carbon black (GCB), NH_2_, and C_18_ with a diameter of 50 μm, were bought from Dikma Technologies Inc. (Beijing, China).

### 2.3. Sample Preparation and Analysis

#### 2.3.1. Sample Preparation Method

The experiment was performed in accordance with the regulations (NY/T 789-2004) established by the Ministry of Agriculture and Rural Affairs of the People’s Republic of China. The Pu’er tea samples (about 300 g) were ground and passed through a 200 mm polyethylene sieve. Two grams of tea leaves was weighed and placed into a 50 mL centrifuge tube. Ten milliliters of water were added, vortexed for half a minute, and allowed to stand for 30 min. After the water in the Pu’er tea was fully soaked, 15 mL of acetonitrile was added and vortexed for 2 min. Then, 3 g sodium chloride was added, vortexed for half a minute and centrifuged at 5000 r/min. Two milliliters of the upper extract were taken out and placed into another 10 mL centrifuge tube. A total of 300 mg of anhydrous magnesium sulfate and 40 mg of multi-walled carbon nanotubes were added, vortexed for half a minute and centrifuged at 5000 r/min. The supernatant was filtered through a 0.22 μm filter membrane before analysis.

#### 2.3.2. Instrumental Analysis Method

Sample analyses were performed with an AB Sciex QTRAP 5500 mass spectrometer (MS/MS) (Framingham, MS, USA) equipped with 1290Ⅱ Infinity UHPLC (Agilent Technology, Santa Clara, CA, USA) and Waters ACQUITY UPLC BEH T3 column (2.1 × 50 mm, 1.8 μm, Waters, MA, USA). Solvents A (1 mM ammonium formate in ultrapure water with 0.1% formic acid) and B (methanol) were used at a flow rate of 0.3 mL/min with the following gradient: 5% B → 95% B (0~3.5 min) → 95% B (3.5~5.5 min) → 5% B (5.5~5.7 min) → 5% B (5.7~8.0 min). The injection volume was 1 µL.

The electrospray ionization (ESI) source was operated in positive (ESI^+^) mode for forming [analyte + H]^+^ ions. Analyte ion transitions used for qualification and quantitation were monitored using multiple reaction monitoring (MRM) mode. Ion source conditions were as follows: ionspray voltage, 5500 V; heating gas temperature, 550 °C; curtain gas flow rate, 20 L/h; nebulizing gas flow rate, 55 L/h; heating gas flow rate, 55 L/h. The identification of proper ion transitions (precursor ion > product ion) of each pesticide and the optimization of a number of MS/MS parameters, including declustering potential (DP) and collision energy (CE), were performed with a syringe pump, providing a constant flow of the standard solution (0.1 μg/mL) of four pesticides to the MS/MS at a flow rate of 10 μL/min. An enhanced product ion (EPI) scan mode was applied in QTrap. EPI mode was acquired from 50 to 600 amu with a scan speed of 10,000 amu/s. The CE and collision energy spread (CES) were 35 V and 15 V, respectively. The other parameters for the detection of the four pesticides are shown in Table 1.

#### 2.3.3. Method Validation

The quantification limits, linear ranges and correlation coefficients of each pesticide in Pu’er teas were determined according to SANTE/ 12682/2019 [31]. According to the actual sensitivity of the instrument for detecting the four pesticides, 0.01–5.0 ng/mL was selected as the linear range of the four pesticides, including 0.1, 0.2, 0.5, 1.0, 2.0 and 5.0 ng/mL. The limit of quantification (LOQ) and limit of detection (LOD), estimated at signal-to-noise ratios of 10 and 3, were evaluated by spiked blank Pu’er tea samples for quality control. Recovery and reproducibility experiments were conducted in 6 replicates, each at 3 concentration levels (LOQs, 5 × LOQs, 10 × LOQs). The intraday and intraday precision were determined by recovering 3 concentration levels (LOQs, 5 × LOQs, 10 × LOQs) at different time points on the same day, and on different days of the week, with 6 replicates for each added concentration. Their stability was assessed by calculating the RSD.

### 2.4. Matrix Effect

The slopes of calibration curves of 4 pesticides in different purification materials and organic solvents were studied to evaluate matrix effects. Matrix effects (ME) were measured according to Equation (1) [32], as follows:*ME* = (slope of solvent standard/slope of matrix matched standard − 1) × 100%.(1)

This means that the matrix is inhibited when ME < 0 and the matrix is enhanced when ME > 0. There is a weak matrix effect when the absolute value of ME is 0–20%, a medium matrix effect when the absolute value of ME is 20–50%, and a strong matrix effect when the absolute value of ME is 50%.

### 2.5. Risk Assessment

The pesticide hazard quotient (HQ) of long-term potential health effects was evaluated as Equation (2). The HQ can be interpreted as follows: HI < 1, the risk is considered acceptable; HI ≥ 1, there is an unacceptable risk [29].
HQ = EDI/ADI(2)
where ADI (mg kg^−1^ bw) is the acceptable daily pesticide intake (the ADI of every pesticide can be found in GB 2763-2021) and EDI is the estimated daily intake, which was calculated as Equation (3) as follows:EDI = C × D × T/Bw(3)
where C is the average residual content of a pesticide in Pu’er tea (mg/kg), D is the amount of tea consumed per day (g), T is the transfer rate of pesticide residue from made tea to tea infusion, and Bw is the body weight (kg).

## 3. Results

### 3.1. Optimization of Instrument Conditions

In this experiment, a T3 chromatographic column, with better separation performance than C_18,_ was selected [33,34], which can complete the analysis of a sample in 8 min. At the same time, the ESI^+^ mode was selected for the determination of four pesticides; the peak shapes and the response of the four pesticides were improved by adding a certain amount of formic acid and ammonium acetate in the mobile phase. The chromatograms of the four pesticides are shown in Figure 1.

### 3.2. QTrap System

EPI (enhanced product ion) mode is a common mode in QTrap scanning. In this mode, the parent ion of the target compound is first selected in the first quadrupole (Q1), and all other ions are filtered out. The parent ions are generated via collisionally activated dissociation (CAD) in the collision cell of the second quadrupole (Q2), and the fragment ions are captured and enriched by the ion trap to obtain an enhanced secondary ion full scan mass spectrum. Under low-concentration conditions, QTrap can be used to obtain high-quality secondary mass spectra. At the same time, the target compound can be confirmed by matching the secondary mass spectra of the standard and positive samples. Figure 2 shows the secondary mass spectra obtained by using QTrap for four pesticides. As shown in Figure A1 (Appendix A), the secondary mass spectra of the positive sample and the standard were compared, and the comparison results show that the matching degree of the four pesticides was greater than 95%, indicating that the use of QTrap can further improve the sensitivity, confirmation and reliability of pesticide detection for Pu’er tea.

### 3.3. Matrix Effects

The slopes of the standard curves prepared by acetonitrile extract, acetonitrile extract after purification with different purification materials (MWCNTs, NH_2_, PSA, florisil, C_18_, and GCB), and methanol have been compared. According to the calculation formula of the matrix effect, the standard curve made of methanol should be used as the benchmark; the closer the slope of other curves is to methanol, the smaller the matrix effect will be. The four pesticides are shown in Figure 3. For chlorpyrifos, the slopes of the GCB and MWCNT purification solutions were closer to methanol. For triazophos, the slopes of the MWCNT and C_18_ purification solutions were closer to methanol. For indoxacarb, the slopes of the MWCNT and PSA purification solutions were relatively close to those of methanol. For tolfenpyrad, the slopes of the MWCNT and C_18_ purification solutions were relatively close to those of methanol. Although florisil, PSA, and C_18_ were effective in reducing the matrix effect of triazophos, indoxacarb, and tolfenpyrad, interfering substances, such as pigments, polyphenols, and acidic components in Pu’er tea, can be effectively removed by MWCNTs, which can effectively reduce the matrix effect of the four pesticides [14,35,36]. Therefore, MWCNT was selected as the suitable purification material for the four pesticides.

### 3.4. Optimization of the Amount of MWCNTs and MgSO_4_

The effects of MWCNT addition (10, 20, 30, 40, 50, and 60 mg) on the recovery of four pesticides were compared. Figure 4 shows that when the amount of MgSO_4_ added was 300 mg, with the increase in the addition of MWCNTs, the recovery rates of chlorpyrifos, tolfenpyrad and triazophos exhibited a decreasing trend. However, when the addition amount was 40 mg, the recovery rates of chlorpyrifos and tolfenpyrad were close to stable, while the recovery rate of indoxacarb was largely unaffected by the addition amount of MWCNTs. Therefore, an addition of 40 mg MWCNTs was considered to be the best choice.

On the other hand, the effects of different amounts of MgSO_4_ (50, 100, 200, 300, 400, 500 mg) on the recovery effects of the four pesticides were also compared. In Figure 5, when the added amount of MWCNTs was 40 mg, as the amount of MgSO_4_ added increases, the recovery rate of chlorpyrifos and tolfenpyrad continues to increase. It is possible that the added amount of MgSO_4_ better absorbs water in the solution environment, improving the extraction rate. Triazophos has little effect on the added amount. The recovery of indoxacarb increases with the increase in the added amount. In general, the addition of too little or too much MgSO_4_ will affect the recovery of the target compound. When the added amount was 200–300 mg, a better recovery effect was obtained. Taking this into consideration, we decided that 300 mg was the best option.

### 3.5. Method Validation

As shown in Table 2, linearity was studied in the range of 0.01–5.0 μg/L for four pesticides by matrix-matched standard calibration with a good linear range. The described method was tested for limits of quantification of the four pesticides. Table 2 shows that the limits of quantitation (LOQs), the limits of detection (LODs), and the matrix effects of the pesticides’ values were in the range of 0.10–0.50 μg/kg and −17.8–13.8, respectively. The LOQs in this study were compared with the LOQs in other studies for these four pesticides in tea, and the results showed that this method has greater detection sensitivity than the methods used in other studies [15,37,38,39].

The recoveries of these four pesticides in Pu’er tea were determined by carrying out six consecutive extractions (*n* = 6) at three concentration levels (LOQs, 5 × LOQs, 10 × LOQs). The values were calculated using blank Pu’er tea matrix-matched calibration standards, as shown in Table 3, which details the recovery and relative standard deviation data for the four pesticides. The recoveries of the four pesticides in Pu’er tea were in the range of 74.8–105.0%, with relative standard deviations (RSDs) in the range of 3.9–6.6%.

### 3.6. Pesticide Residues in Pu’er Tea

According to Table 4, the detection rate of tolfenpyrad among the four pesticides was relatively high (35.7%), and the lowest was indoxacarb (5.2%). By comparing these data with China’s pesticide residue limits in tea (GB 2763-2021), the over standard rates of 25 chlorpyrifos, tolfenpyrads, and indoxacarbs were 4.35%, 0.87%, and 0%, respectively. Because different countries have different pesticide residue limits (Table 5) [40], four pesticides in Pu’er tea that do not meet the national standards of other nations will not be discussed in this study. However, by comparing these limit standards with those of other countries and regions, especially Japan, England, and the European Union, which import large amounts of Pu’er tea from China, it is clear that special attention should be paid to their limit standards, as they are stricter than those in China.

### 3.7. Consumer Exposure Assessment

According to Table 6, the HQ of the four pesticides was far less than one, which indicates that the risk of these four pesticides in Pu’er tea was considered acceptable. The HQ of these pesticides ranges from 0.000757 to 0.0135, of which the highest HQ was tolfenpyrad, with the highest detection rate, while the smallest was indoxacarb, with the lowest detection rate.

Although it was acceptable to evaluate the risk of Pu’er tea using the HQ value of tolfenpyrad, its low ADI value (0.006) [13] indicates that its allowable daily intake was low. Considering this, and the determined results of 300 samples, in which the detection rate, mean residue level, and median residue level of tolfenpyrad were all at high levels, subsequent attention should be paid to its presence within our diets in the future.

## 4. Conclusions

In this study, four pesticides with high detection rates in Yunnan Pu’er tea were determined using optimized selection of MWCNTs, combined with QuEChERS-UHPLC/QTRAP-MS/MS methods, and the effects of different purification methods on their matrix effects were compared. The LOQs for the four pesticides ranged from 0.10 to 0.50 μg/kg, and the matrix effect ranged from –17.8 to 13.8. The recoveries of the four pesticides in Pu’er tea were in the range of 74.8–105.0%, with the RSDs in the range of 3.9–6.6%. The presence of the four pesticides in Pu’er tea, with a complex matrix, was strongly confirmed using QTRAP technology. This method was applied to samples taken from three areas in Yunnan in which Pu’er tea is commonly produced, in order to determine the presence of these four pesticides. The results show that the highest detection rate was that of tolfenpyrad, 35.7%, while the lowest was indoxacarb, 5.2%. The HQ of these four pesticides was far less than one, indicating that the risk was considered acceptable for the four pesticides in Pu’er tea. However, while it was acceptable to evaluate its risk using the HQ value of tolfenpyrad, according to the test results, the detection rate, average residual value and residual median value of tolfenpyrad were all at relatively high levels, and attention should be paid to its presence within the public’s diet in the future.

## Figures and Tables

**Figure 1 molecules-27-01053-f001:**
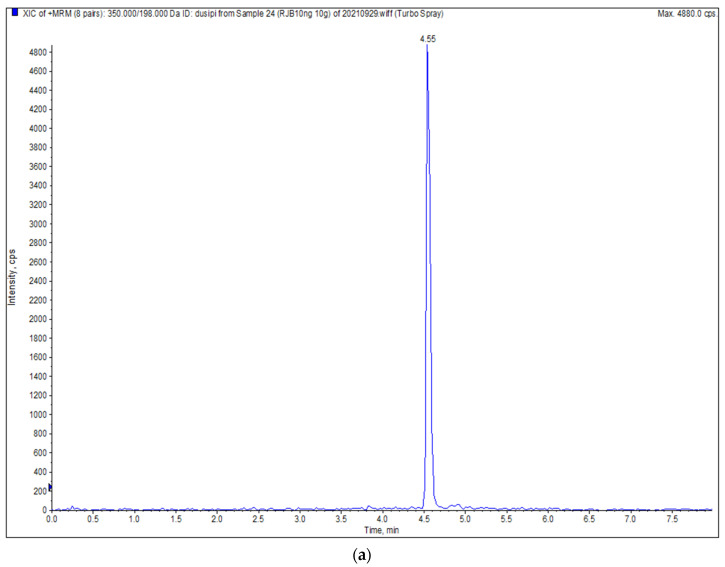
Chromatograms of 4 pesticides. (**a**) Chlorpyrifos; (**b**) triazophos; (**c**) tolfenpyrad; (**d**) indoxacarb.

**Figure 2 molecules-27-01053-f002:**
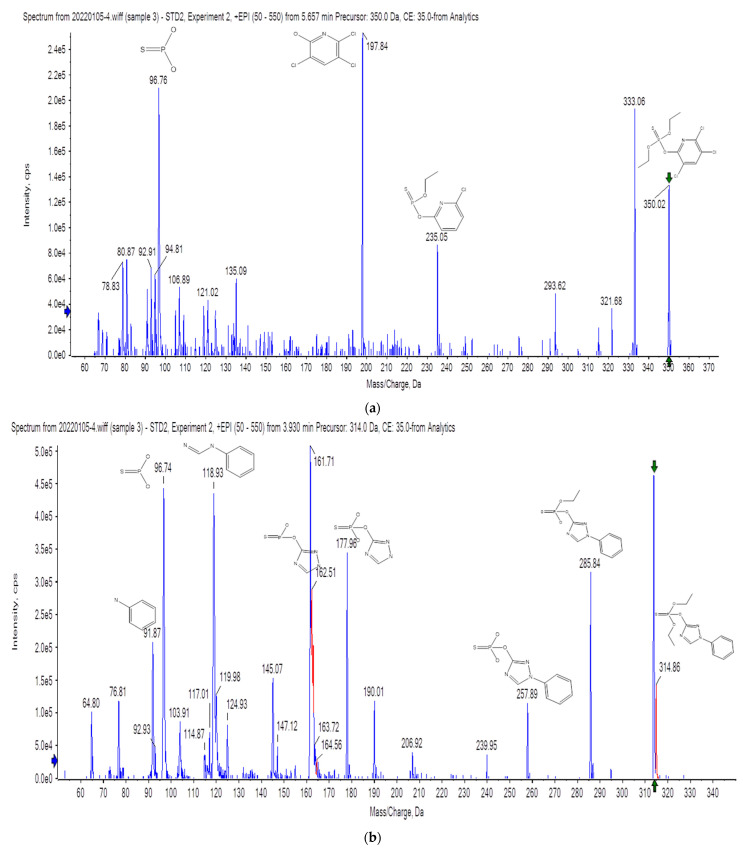
QTrap spectrum (EPI) of positive samples and standard for four pesticides in Pu’er tea. (**a**) Chlorpyrifos; (**b**) triazophos; (**c**) tolfenpyrad; (**d**) indoxacarb.

**Figure 3 molecules-27-01053-f003:**
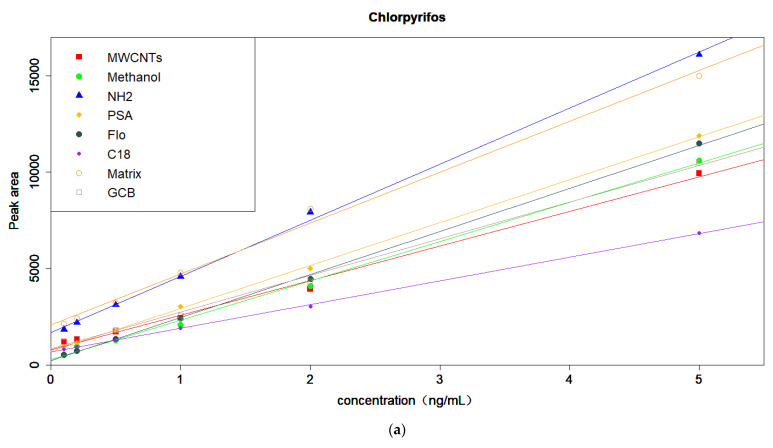
Standard curve of matrix preparation with different preparation methods. (**a**) Chlorpyrifos; (**b**) triazophos; (**c**) tolfenpyrad; (**d**) indoxacarb.

**Figure 4 molecules-27-01053-f004:**
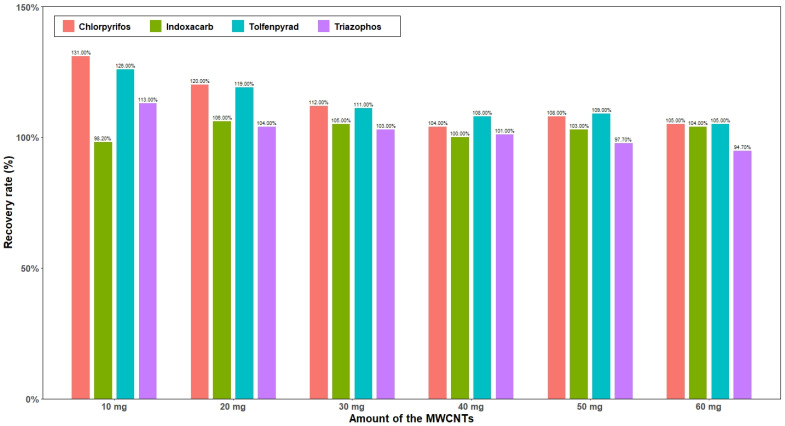
The influence of different contents of MWCNTs on the recovery rate of four pesticides.

**Figure 5 molecules-27-01053-f005:**
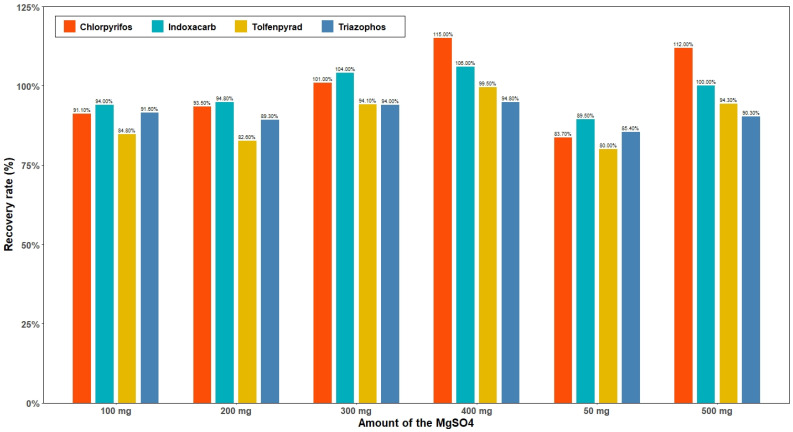
The influence of different contents of MgSO_4_ on the recovery rate of four pesticides.

**Table 1 molecules-27-01053-t001:** UHPLC-MS/MS parameters for detection of the four pesticides.

Pesticide	Ionization Mode	Precursor Ion/(*m*/*z*)	Product Ion/(*m*/*z*)	DP/(Volts)	CE/(Volts)
Chlorpyrifos	ESI^+^	350.0	198.0 */96.9 ^#^	60	25/40
Triazophos	ESI^+^	314.0	162.0 */119.1 ^#^	55	25/50
Tolfenpyrad	ESI^+^	384.0	197.1 */154.1 ^#^	60	35/55
Indoxacarb	ESI^+^	528.0	249.1 */293.1 ^#^	60	25/22

* quantitative ion, # qualitative ion.

**Table 2 molecules-27-01053-t002:** Linear ranges, correlation coefficients (*R^2^*), limits of quantitation (LOQs), limit of detection (LODs) and matrix effect of four pesticides.

Pesticide	Linear Ranges (μg/L)	*R^2^*	LOQ (μg/kg)	LOD (μg/kg)	ME%
Chlorpyrifos	0.01–5.0	0.9944	0.10	0.03	13.8
Triazophos	0.01–5.0	0.9997	0.50	0.15	−17.8
Tolfenpyrad	0.01–5.0	0.9996	0.20	0.06	−5.8
Indoxacarb	0.01–5.0	0.9995	0.50	0.15	7.3

**Table 3 molecules-27-01053-t003:** Recoveries, relative standard deviations, intraday and interday precisions of four pesticides.

Pesticide	Spiked Amount (μg/kg)	Recovery (%)/RSD (%)	Intraday Precision (%)	Interday Precision (%)
Chlorpyrifos	0.10	85.4/5.9	4.5	5.2
	0.50	103.0/6.4	4.2	5.8
	1.00	105.0/5.3	3.7	4.7
Triazophos	0.50	74.8/5.3	3.7	4.1
	2.50	92.9/4.7	4.3	4.5
	5.00	96.4/3.9	3.9	3.5
Tolfenpyrad	0.20	81.3/6.5	4.9	5.6
	1.00	97.3/6.6	4.7	3.5
	2.00	101.0/5.9	3.9	4.5
Indoxacarb	0.50	90.4/6.7	4.7	3.6
	2.50	98.1/5.8	3.5	4.4
	5.00	96.4/4.2	3.6	4.6

**Table 4 molecules-27-01053-t004:** Pesticide residues detected in Pu’er tea.

Pesticide	Detection Rate (%)	Range of Detected Content (mg/kg)	Mean Residue Level (mg/kg)	Median Residue Level (mg/kg)
Chlorpyrifos	12.2	1.10–5.28	2.14	1.62
Triazophos	10.4	0.014–0.103	0.049	0.046
Tolfenpyrad	35.7	1.02–51.8	11.6	5.01
Indoxacarb	5.2	1.07–4.89	2.84	2.96

**Table 5 molecules-27-01053-t005:** The maximum residue limits for four pesticides in tea in different countries and regions.

Pesticide	China (mg/kg)	England (mg/kg)	Japan (mg/kg)	Korea (mg/kg)	European Union (mg/kg)	CAC * (mg/kg)	Canada (mg/kg)	America (mg/kg)
Chlorpyrifos	2	0.10	10	2.0	2.0	2.0	—	—
Triazophos	—	0.02	0.05	0.02	0.02	—	—	—
Tolfenpyrad	50	—	20	30		30	30	30
Indoxacarb	5	—	—	—	0.05	5	—	—

* CAC: Codex Alimentarius Commission.

**Table 6 molecules-27-01053-t006:** Consumer exposure assessment of 4 pesticides in Pu’er tea.

Pesticide	C (mg/kg)	D ^a^ (g)	T (%)	Bw ^b^	EDI	ADI (mg/kg bw) [13]	HQ
Chlorpyrifos	2.14	10	8.6 [41]	60	3.07 × 10^−5^	0.01	0.00307
Triazophos	0.049	10	27.1 [42]	60	2.21 × 10^−6^	0.001	0.00221
Tolfenpyrad	11.6	10	4.2 [41]	60	8.12 × 10^−5^	0.006	0.0135
Indoxacarb	2.84	10	1.6 [43]	60	7.57 × 10^−6^	0.01	0.000757

^a^ Daily intake of Tieguanyin tea is 10 g and ^b^ the adult body weight is 60 kg [44].

## Data Availability

Data are contained within the article.

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
