# Peer review of "Simultaneous Determination and Health Risk Assessment of Four High Detection Rate Pesticide Residues in Pu’er Tea from Yunnan, China"

_molecules, 2022, doi:10.3390/molecules27031053_

Round 1

Reviewer 1 Report

Reviewer’s comments

The article entitled “Simultaneous determination and health risk assessment of four high detection rate pesticide residues in Pu’er tea from Yunnan, China” describes optimization and validation of QuEChERS method for extraction of four pesticides in tea and analysis by UHPLC-QTRAP-MS/MS. In addition, the authors briefly showed a consumer exposure assessment.

The article is poorly written, mainly the item "3.3. Matrix effects" and "3.4. Optimization of the amount of the MWCNTs and MgSO4”. The authors did not present an appropriate discussion of the results.

The Figure 1 is very large.

The article has excess of figures.

Page 2 lines 46-48 and 49-50: There is repeated information.

Page 12 line 232-233. MgSO4 was used during d-SPE. In this step, the extracts are cleaned by sorbent and remaining water is removed by MgSO4. Thus, in this step there is no liquid-liquid extraction.

Table 2. The first level of the curve is less than LOQ. This is not allowed.

LOQ is 0.10 μg/kg corresponding to 0.0133 μg/L in the extract before analysis. 

Author Response

Point 1: The article is poorly written, mainly the item "3.3. Matrix effects" and "3.4. Optimization of the amount of the MWCNTs and MgSO4”. The authors did not present an appropriate discussion of the results.

Response 1: Dear reviewer, in section 3.3, the slopes of the standard curves prepared by different purification materials were compared and reflected in the figure, and the effect of different additions of MWCNTs and MgSO4 on the recovery effect was also presented in the figure. The corresponding content has been supplemented in the text.

Point 2: The Figure 1 is very large.

Response 2: Dear reviewer, figure 1 has been modified to follow the journal's format.

Point 3: The article has excess of figures.

Response 3: Dear reviewer, some figures have been moved to the appendix A.

Point 4: Page 2 lines 46-48 and 49-50: There is repeated information.

Response 4: The duplicate content were revised.

Point 5: Page 12 line 232-233. MgSO4 was used during d-SPE. In this step, the extracts are cleaned by sorbent and remaining water is removed by MgSO4. Thus, in this step there is no liquid-liquid extraction.

Response 5: Dear reviewer, sorry for our writing mistakes, the mistakes have been revised.

Point 6: Table 2. The first level of the curve is less than LOQ. This is not allowed.

Response 6: In this experiment, the research method of the limit of quantification was as follows: after adding a certain concentration of standard to the blank Pu'er tea tea matrix, it was carried out according to the pretreatment method of this experiment. Calculated according to the limit of quantification of 0.1 μg/kg, the sample weight was 2 g, and 15 mL of extraction solvent was added. The instrument-determined concentration corresponding to the limit of quantification should be 0.0133 ng/mL, which should be within the linear range of the pesticide.

Point 7: LOQ is 0.10 μg/kg corresponding to 0.0133 μg/L in the extract before analysis. 

Response 7: Dear reviewer, the corresponding explanation is the same as Response 6. After the tea samples were pre-treated, the concentration of the instrument should be within the linear range.

Reviewer 2 Report

The authors describes an analytical method to determine the amount of four pesticides, namely Chloropyrifos, Triazophos, Tolfenpyrad and Indoxacarb that can be found in Chinese tea. Pesticide are extracted using multiwalled carbon nanotube extraction technique. Other consideration about health risk assessment and estimated daily intake are also reported.

In my opinion, significant improvement are necessary to accept the manuscript for publication.  The manuscript is poor written and an extensive language revision is required. I am listing here some of the point that should be changed/rephrased for a better understanding.

Abstract

Line 15: Use the full form for QuEChERS before the acronym

Line 19: do not use fine, rephrase.

Line 20: change “with the ranged from” with “values were in the range”.

Line 23: rephrase and insert the number of samples instead of the “actual sample”.

Line 25: what is the over standard rates?

Line 27: add “obtained” between “results” and “with”.

Line 28: use “is acceptable” instead of was.

Lines 30-31: this sentence is unclear, please rephrase.

Introduction

Line 50-52: rephrase.

Line 55: the use of “more” is incorrect. Toluene is more toxic compared to what?. Please remove o rephrase.

Lines 59-63: author statements are not correct and are not supported by any reference.  Rephrase or add reference.

Lines 64-69: rephrase.

Line 77: unclear, rephrase.

Material and methods

Line 91: add a brief explanation of the sampling method.

Line 93: What is the solvent used for the 1000 ppm standard solution preparation?

Line 114: change with “was filtered … before analysis”.

Line 122: remove “in both” if only positive mode was used.

Table1: please list qualifier and quantifier.

Line 138: according to this reviewer knowledge, the most recent guideline for pesticides is the SANTE/12682/2019. Please, verify and change the reference.

Lines 139-141: add information about linearity range tested. Method for LOQ calculation, number of point for the calibration curves, number of replicates (intraday and interday), add LOD.

Line 143-145: Unclear, rephrase.

Results

Figure 1: caption is insufficient. Please insert a,b,c,d (or 1,2,3,4) in the chromatogram and the corresponding pesticide in the caption, please add also the analytical condition.   

Figure 2: caption is insufficient. Please insert a,b,c,d (or 1,2,3,4)  in the chromatogram and the corresponding pesticide in the caption. Figures are blurry and stretched. Provide a better quality ones.

Figure 3: same consideration as above.

Line 199: insert the same reference reported in material method.  This section should be completely modified. The authors should discuss better why they decided to use MWCNT, what are the advantages, list the common method present in literature and used for extraction of pesticides, and add some reference to them.

Line 220-226: please rephrase.

Lines: 241-244: please avoid repeating “for the four pesticide” and rephrase accordingly.

Lines 244-245: change “with the ranged from” with “values were in the range”.

Table 2: use the same unit for the linear range and LOQ, add column and values for LOD. 

Line 250: explain what matrix was used for the matrix matched calibration.

Table 3. remove “Added” and use “spiked amount” instead.

Lines 267-275: rephrase

Table 6: add a column for the ADI values reported in GB 2763-2021.

Conclusions

Please rephrase the text using the same correction provided for abstract.

Author Response

Point 1: Line 15: Use the full form for QuEChERS before the acronym

Response 1: Dear reviewer, QuEChERS has been revised.

Point 2: Line 19: do not use fine, rephrase.

Response 2: Dear reviewer, “fine” has been revised.

Point 3: Line 20: change “with the ranged from” with “values were in the range”.

Response 3: Dear reviewer, it has been replaced.

Point 4: Line 23: rephrase and insert the number of samples instead of the “actual sample”.

Response 4: Dear reviewer, it has been inserted.

Point 5: Line 25: what is the over standard rates?

Response 5: Dear reviewer, the over standard rates were compared with China's pesticide residue limits in tea, related content has been added.

Point 6: Line 27: add “obtained” between “results” and “with”.

Response 6: Dear reviewer, it has been added.

Point 7: Line 28: use “is acceptable” instead of was.

Response 7: Dear reviewer, “was” has been replaced with “is”.

Point 8: Lines 30-31: this sentence is unclear, please rephrase.

Response 8: Dear reviewer, it has been revised.

Point 9: Line 50-52: rephrase.

Response 9: Dear reviewer, it has been revised.

Point 10: Line 55: the use of “more” is incorrect. Toluene is more toxic compared to what?. Please remove or rephrase.

Response 10: Thank you, “more” has been removed.

Point 11: Lines 59-63: author statements are not correct and are not supported by any reference.  Rephrase or add reference.

Response 11: Dear reviewer, it has been revised.

Point 12: Lines 64-69: rephrase.

Response 12: Dear reviewer, it has been revised.

Point 13: Line 77: unclear, rephrase.

Response 13: Dear reviewer, it has been revised.

Point 14: Line 91: add a brief explanation of the sampling method.

Response 14: Dear reviewer, it has been added.

Point 15: Line 93: What is the solvent used for the 1000 ppm standard solution preparation?

Response 15: Dear reviewer, it has been added.

Point 16: Line 114: change with “was filtered … before analysis”.

Response 16: Dear reviewer, it has been revised.

Point 17: Line 122: remove “in both” if only positive mode was used.

Response 17: Dear reviewer, sorry for our mistakes in writing, it has been revised.

Point 18: Table1: please list qualifier and quantifier.

Response 18: Dear reviewer, it has been revised.

Point 19: Line 138: according to this reviewer knowledge, the most recent guideline for pesticides is the SANTE/12682/2019. Please, verify and change the reference.

Response 19: Dear reviewer, it has been revised.

Point 20: Lines 139-141: add information about linearity range tested. Method for LOQ calculation, number of point for the calibration curves, number of replicates (intraday and interday), add LOD.

Response 20: Dear reviewer, the corresponding contents have been added.

Point 21: Line 143-145: Unclear, rephrase.

Response 21: Dear reviewer, it has been revised.

Point 22: Figure 1: caption is insufficient. Please insert a,b,c,d (or 1,2,3,4) in the chromatogram and the corresponding pesticide in the caption, please add also the analytical condition.

Response 22: Dear reviewer, relevant figures have been modified according to the journal format.

Point 23: Figure 2: caption is insufficient. Please insert a,b,c,d (or 1,2,3,4)  in the chromatogram and the corresponding pesticide in the caption. Figures are blurry and stretched. Provide a better quality ones.

Response 23: Dear reviewer, relevant figures have been modified according to the journal format.

Point 24: Figure 3: same consideration as above.

Response 24: Dear reviewer, relevant figures have been modified according to the journal format.

Point 25: Line 199: insert the same reference reported in material method.  This section should be completely modified. The authors should discuss better why they decided to use MWCNT, what are the advantages, list the common method present in literature and used for extraction of pesticides, and add some reference to them.

Response 25: Dear reviewer, it has been revised.

Point 26: Line 220-226: please rephrase.

Response 26: Dear reviewer, it has been revised.

Point 27: Lines: 241-244: please avoid repeating “for the four pesticide” and rephrase accordingly.

Response 27: Dear reviewer, it has been revised.

Point 28: Lines 244-245: change “with the ranged from” with “values were in the range”.

Response 28: Dear reviewer, it has been revised. Thank you.

Point 29: Table 2: use the same unit for the linear range and LOQ, add column and values for LOD. 

Response 29: Dear reviewer, it has been revised. Thank you.

Point 30: Line 250: explain what matrix was used for the matrix matched calibration.

Response 30: Dear reviewer, blank Pu'er tea matrix were added.

Point 31: Table 3. remove “Added” and use “spiked amount” instead.

Response 31: Dear reviewer, it has been revised. Thank you.

Point 32: Lines 267-275: rephrase

Response 32: Dear reviewer, it has been revised.

Point 33: Table 6: add a column for the ADI values reported in GB 2763-2021.

Response 33: Dear reviewer, it has been revised.

Point 34: Please rephrase the text using the same correction provided for abstract.

Response 34: Dear reviewer, it has been revised. Thank you.

Reviewer 3 Report

The manuscript (molecules-1569084) titled “Simultaneous determination and health risk assessment of four high detection rate pesticide residues in Pu'er tea from Yunnan, China” by Tao Lin, Xing-Lian Chen, Jin Guo, Meng-Xia Li, Yu-Feng Tang, Mao-xuan Li, Yan-gang Li, Long Cheng and Hong-Cheng Liu describes the optimization of the method for determination of four pesticides in tea using UHPLC-QTRAP-MS/MS and modified QuEChERS protocol. The method is fast and can be used for screening pesticide residues in this kind of material, what is very important.    

I have some comments which could influence on the value of the manuscript and can improve their quality:

  1. Whether the QuEChERS method or other analytical methods were used in the determination of pesticides in tea? If yes, please discuss it in the introduction. Have an in-depth discussion of pesticide determination methods in tea. (line 53-62 page 2).
  2. Lack of discussion yours research with the results of other authors in “Results”. Please add it.
  3. Line 43 page 1 - Tolfenpyrad and indoxacarb are commonly used pesticides in Pu'er tea – is this sentence correct. I think that the more correct form of this sentence is: Tolfenpyrad and indoxacarb are commonly used pesticides for Pu'er tea production or during tea plant
  4. Why were only four compounds selected? Where did the authors get the data on chlorpyrifos, triazophos, tolfenpyrad and indoxacarb as high detection rate pesticides? Lack of references.
  5. Are other pesticides also detected in tea samples? Perhaps it would be worth describing this fact in introduction. As literature proposals: (i) Qinghua Yao, Sun-An Yan, Jie Li, Minmin Huang & Qiu Lin (2020) Health risk assessment of 42 pesticide residues in Tieguanyin tea from Fujian, China, Drug and Chemical Toxicology, DOI: 1080/01480545.2020.1802476; (ii) Hongping Chen , Qinghua Wang, Ying Jiang, Chuanpi Wang, Peng Yin, Xin Liu, Chengyin Lu. Monitoring and risk assessment of 74 pesticide residues in Pu-erh tea produced in Yunnan, China. Food Addit Contam Part B, 2015; 8(1):56-62. doi: 10.1080/19393210.2014.972471. In the future, it may be worth considering extending the QuEChERS method to include new compounds.
  6. Line 45-49 page 2 “Chlorpyrifos and triazophos are also high detection rate pesticides. They are currently banned in China due to their high rate of exceeding the standard in vegetables and their moderate toxicity, chlorpyrifos and triazophos are also high detection rate pesticides . At present, they are banned in China due to their high rate of exceeding the standard in vegetables and their moderate toxicity”. A few of information are repeated in these sentences. Please, change it.
  7. Line 60 page 2 – should be “high” instead of “High”?
  8. Line 69 page 2 – should be “Linear” instead of Iinear”?
  9. Line 138 page 3, Why the method validation was performed according to the SANTE guide (SANTE/11813/2017). New edition available: SANTE/12682/2019.
  10. In addition to the estimated validation parameters, the uncertainty of the method should also be calculated.
  11. Line 143-144 page 4 – that sentence is not understand for me. Please, change it.
  12. Figure 1 – is it possible to present all analyzed compounds in one chromatogram?
  13. Could the authors explain why they used such a sorbent for purification step? and describe what interfering substances in the tea matrix are removed by these sorbents.
  14. Table 5 – lack of references for all documents in which MRLs are available.
  15. The abbreviation of RSDs (line 285 page 14) was explain for the first time in line 253 page 13. Please, remove it.
  16. Not all abbreviations are explain, for example CAC (table 5). Please, check all manuscript.
  17. Please standardize the units in all manuscript, for example: line 130, page 3 is μg mL-1 while in line 244 page 12 is μg/kg.

After the corrections I recommend this paper to publication.

Author Response

Point 1: Whether the QuEChERS method or other analytical methods were used in the determination of pesticides in tea? If yes, please discuss it in the introduction. Have an in-depth discussion of pesticide determination methods in tea. (line 53-62 page 2).

Response 1: Dear reviewer, it has been revised.

Point 2: Lack of discussion yours research with the results of other authors in “Results”. Please add it.

Response 2: Dear reviewer, the relevant content has been supplemented in “3.3. Matrix effects” and “3.5. Method Validation”.

Point 3: Line 43 page 1 - Tolfenpyrad and indoxacarb are commonly used pesticides in Pu'er tea – is this sentence correct. I think that the more correct form of this sentence is: Tolfenpyrad and indoxacarb are commonly used pesticides for Pu'er tea production or during tea plant

Response 3: Dear reviewer, thanks for pointing out the mistakes, it has been revised.

Point 4: Why were only four compounds selected? Where did the authors get the data on chlorpyrifos, triazophos, tolfenpyrad and indoxacarb as high detection rate pesticides? Lack of references.

Response 4: Dear reviewer, Pu'er tea is one of the famous tea varieties. China's Pu'er tea has a large export volume every year, and China's Yunnan Province is the main producing area of Pu'er tea. Every year, a large number of Pu'er tea samples in Yunnan Province are tested for multi-residue by our institution. Based on our monitoring data on pesticides in Pu'er tea for many years, we have obtained that these four pesticides are the pesticides with high detection rate in Pu'er tea.

Point 5: Are other pesticides also detected in tea samples? Perhaps it would be worth describing this fact in introduction. As literature proposals: (i) Qinghua Yao, Sun-An Yan, Jie Li, Minmin Huang & Qiu Lin (2020) Health risk assessment of 42 pesticide residues in Tieguanyin tea from Fujian, China, Drug and Chemical Toxicology, DOI: 1080/01480545.2020.1802476; (ii) Hongping Chen , Qinghua Wang, Ying Jiang, Chuanpi Wang, Peng Yin, Xin Liu, Chengyin Lu. Monitoring and risk assessment of 74 pesticide residues in Pu-erh tea produced in Yunnan, China. Food Addit Contam Part B, 2015; 8(1):56-62. doi: 10.1080/19393210.2014.972471. In the future, it may be worth considering extending the QuEChERS method to include new compounds.

Response 5: Dear reviewer, Yao's literature, which mainly studies pesticide residues in Tieguanyin tea, is different from the Pu'er tea studied in this paper in terms of planting methods and planting environment. Therefore, this literature has little reference for the determination of pesticides in this paper. According to Hongping Chen's literature, combined with the results of our laboratory's determination of pesticide residues in Pu'er tea in the past 3 years, pesticides such as imidacloprid, bifenthrin and acetamiprid were also detected in our laboratory, but our detection rate is low, combined with the biological toxicity and ADI of imidacloprid, bifenthrin and acetamiprid, 4 pesticides such as chlorpyrifos, triazophos, tolfenpyrad and indoxacarb were finally selected as the target pesticides for detection. The relevant content has been modified in introduction. Thank you for your suggestions for future extensions to the QuEChERS method.

Point 6: Line 45-49 page 2 “Chlorpyrifos and triazophos are also high detection rate pesticides. They are currently banned in China due to their high rate of exceeding the standard in vegetables and their moderate toxicity, chlorpyrifos and triazophos are also high detection rate pesticides . At present, they are banned in China due to their high rate of exceeding the standard in vegetables and their moderate toxicity”. A few of information are repeated in these sentences. Please, change it.

Response 6: Dear reviewer, thanks for pointing out the mistakes, it has been revised.

Point 7: Line 60 page 2 – should be “high” instead of “High”?

Response 7: Dear reviewer, it has been revised.

Point 8: Line 69 page 2 – should be “Linear” instead of Iinear”?

Response 8: Dear reviewer, it has been revised.

Point 9: Line 138 page 3, Why the method validation was performed according to the SANTE guide (SANTE/11813/2017). New edition available: SANTE/12682/2019.

Response 9: Dear reviewer, it has been revised.

Point 10: In addition to the estimated validation parameters, the uncertainty of the method should also be calculated.

Response 10: Dear reviewer, we apologize for the lack of the uncertainty of the method in our experimental design. However, according to the two related literatures in point 5 (Qinghua Yao, et al., 2020; Hongping Chen, et al., 2015) and the latest literature on detection methods ((i)Pallavi, M. S., et al. "Simultaneous determination, dissipation and decontamination of fungicides applied on cabbage using LC-MS/MS." Food Chemistry 355 (2021): 129523; (ii) Kecojević, Isidora, et al. "Evaluation of LC-MS/MS methodology for determination of 179 multi-class pesticides in cabbage and rice by modified QuEChERS extraction." Food Control 123 (2021): 107693; (iii) Yang, Fei, et al. "A rapid method for the simultaneous stereoselective determination of the triazole fungicides in tobacco by supercritical fluid chromatography-tandem mass spectrometry combined with pass-through cleanup." Journal of Chromatography A 1642 (2021): 462040.), there are few literatures with uncertain calculation methods. According to our opinion, uncertainty of the method may not need to be added, but if it does need to be added, please let us know in time and we will add it as soon as possible.

Point 11: Line 143-144 page 4 – that sentence is not understand for me. Please, change it.

Response 11: Dear reviewer, it has been revised.

Point 12: Figure 1 – is it possible to present all analyzed compounds in one chromatogram?

Response 12: Dear reviewer, figure 1 has been modified to follow the journal's format.

Point 13: Could the authors explain why they used such a sorbent for purification step? and describe what interfering substances in the tea matrix are removed by these sorbents.

Response 13: Dear reviewer, the main sorbent used in this experiment is multi-walled carbon nanotubes, and sodium chloride and anhydrous magnesium sulfate have little effect on the purification effect in this experiment (the role of sodium chloride is mainly to saturate the aqueous solution and promote the rapid stratification of the acetonitrile layer and the aqueous solution layer, the role of anhydrous magnesium sulfate is mainly to adsorb the water in the solution), therefore, in this experiment, multi-walled carbon nanotubes and other common purification sorbents such as PSA, C18, GCB, NH2 were mainly used. After comparing the adsorption effect, it was finally determined that the purification effect of multi-walled carbon nanotubes was the best, so it was used as the preferred sorbent in this experiment. According to our knowledge, multi-walled carbon nanotubes can effectively remove pigments, polyphenols and acidic components in Pu'er tea.

Point 14: Table 5 – lack of references for all documents in which MRLs are available.

Response 14: Dear reviewer, It was added in section 3.6.

Point 15: The abbreviation of RSDs (line 285 page 14) was explain for the first time in line 253 page 13. Please, remove it.

Response 15: Dear reviewer, it has been revised.

Point 16: Not all abbreviations are explain, for example CAC (table 5). Please, check all manuscript.

Response 16: Dear reviewer, it has been revised.

Point 17: Please standardize the units in all manuscript, for example: line 130, page 3 is μg mL-1 while in line 244 page 12 is μg/kg.

Response 17: Dear reviewer, it has been revised.

Reviewer 4 Report

The paper “Simultaneous determination and health risk assessment of four high detection rate pesticide residues in Pu'er tea from Yunnan, China” is good but it can not be published as it is. The authors need to improve the manuscript and underline the novelty of the paper.

  1. The introduction must be improved with deeper discussion of the state of the art and showing some data to compare with the present work.
  2. Line 64-68 Please, rewrite this sentence clearer and more understandable.
  3. Which is the chosen amount of MWCNTs? It is not clear.
  4. When the different amount of MgSO4 were tested, the quantity of MWCNTs was constant? Which was the amount?
  5. The novelty of the work is not clear. Rewrite the discussion to highlight the potential of the work.
  6. Why only 4 pesticides are detected? They are the only one present in this work or the method can not be used for the detection of other pesticides?
  7. All figures must be done with better resolution.
  8. In the method validation LOD and precision are missing.

Author Response

Point 1: The introduction must be improved with deeper discussion of the state of the art and showing some data to compare with the present work.

Response 1: Dear reviewer, apologies for the incompleteness of the introduction to our article. It has been revised.

Point 2: Line 64-68 Please, rewrite this sentence clearer and more understandable.

Response 2: Apologies for the carelessness of our writing. It has been revised.

Point 3: Which is the chosen amount of MWCNTs? It is not clear.

Response 3: Apologies for the carelessness of our writing, it has been revised in section 3.4.

Point 4: When the different amount of MgSO4 were tested, the quantity of MWCNTs was constant? Which was the amount?

Response 4: Apologies for the carelessness of our writing, it has been revised in section 3.4.

Point 5: The novelty of the work is not clear. Rewrite the discussion to highlight the potential of the work.

Response 5: Dear reviewer, it has been revised in “Introduction”.

Point 6: Why only 4 pesticides are detected? They are the only one present in this work or the method can not be used for the detection of other pesticides?

Response 6: Dear reviewer, Pu'er tea is one of the famous tea varieties. China's Pu'er tea has a large export volume every year, and China's Yunnan Province is the main producing area of Pu'er tea. Every year, a large number of Pu'er tea samples in Yunnan Province are tested for multi-residue by our institution. Based on our monitoring data on pesticides in Pu'er tea for many years, we have obtained that these four pesticides are the pesticides with high detection rate in Pu'er tea. This method is also applicable to the determination of other moderately polar pesticides in Pu'er tea, but the detection rate of other pesticides is low.

Point 7: All figures must be done with better resolution.

Response 7: Dear reviewer, all figures have been modified to follow the journal's format.

Point 8: In the method validation LOD and precision are missing.

Response 8: Dear reviewer, it has been revised.

Reviewer 5 Report

The authors' research is based on the determination of Four pesticides with high detection rate in Pu'er tea. The authors used QuEChERS with multiwalled carbon nanotubes (MWCNs), combined ultrahigh performance liquid chromatography-triple quadrupole linear ion trap-tandem mass spectrometry (UHPLC-QTRAP-MS/MS) method. Additionally, the MWCN material used was compared to other common purification materials.

The manuscript is generally well written. However, the following issues have to be addressed before this manuscript is suitable for publication.

  1. INTRODUCTION

Line: 36-38 Please complete this sentence with references for each effect. It is imperative that you use a bibliographic reference which provides conclusive information; the two references used do not provide information about these relationships. Additionally, please use source citation.

Line: 38-40 Please use bibliographic references that can support this statement. How high is the consumption of Pu'er tea?

Line: 40-42 Please this statement needs a bibliographic reference.

Line: 46-50 Please this statement needs a bibliographic reference.

  1. MATERIALS AND METHODS

Line: 94 Please remove the duplicate word.

Line: 107-110 Sentence too long.

Line: 105-114 Please check the paragraph. Words with a capital letter in the middle of the sentence appear.

  1. RESULTS

In Figure 1, pesticide name captions are missing.

If it possible, please give Figures 2, 4, 5 and 6 of better quality (more readable).

Please check the values of added pesticide in Table 3; especially for: Triazophos and Indoxacarb

Table 5: Please provide references for the standards in the countries listed.

Line 272: Please give literature reference for ADI.

Table 6: Please provide references for ADI for pesticides (below the table).

The manuscript has a high analytical value and the work put into developing the method used should be appreciated. Due to the amount of research material collected, I think it would be worthwhile to mention a few basic issues related to the origin of the collected tea samples:

  1. Did Pu'er tea samples from three different regions show differences in pesticide content?
  2. Which regions of the Pu'er tea samples had a higher pesticide content?

I appreciate the interest of the authors in the development of this manuscript. It is an interesting topic. The work should be redrafted and supplemented with important aspects, therefore I suggest a major revision.

Author Response

Point 1: Line: 36-38 Please complete this sentence with references for each effect. It is imperative that you use a bibliographic reference which provides conclusive information; the two references used do not provide information about these relationships. Additionally, please use source citation.

Response 1: Dear reviewer, it has been revised.

Point 2: Line: 38-40 Please use bibliographic references that can support this statement. How high is the consumption of Pu'er tea?

Response 2: Dear reviewer, according to literature (Dou, Z.; Ji, M.; Wang, M.; Li, H. Empirical analysis of Pu'er tea price bubble measurement based on GSADF method. Acta Agriculturae Scandinavica, Section B — Soil & Plant Science 2021, 71, 81-90), China’s Pu’er tea has reached 170,000 tons in 2018 which accounts for 6.69% of the country’s total tea output. It has been revised.

Point 3: Line: 40-42 Please this statement needs a bibliographic reference.

Response 3: Dear reviewer, it has been revised.

Point 4: Line: 46-50 Please this statement needs a bibliographic reference.

Response 4: Dear reviewer, it has been revised.

Point 5: Line: 94 Please remove the duplicate word.

Response 5: Dear reviewer, it has been revised.

Point 6: Line: 107-110 Sentence too long.

Response 6: Dear reviewer, it has been revised.

Point 7: Line: 105-114 Please check the paragraph. Words with a capital letter in the middle of the sentence appear.

Response 7: Dear reviewer, sorry for not being serious about our writing, it has been revised.

Point 8: In Figure 1, pesticide name captions are missing.

Response 8: Dear reviewer, figure 1 has been modified to follow the journal's format.

Point 9: If it possible, please give Figures 2, 4, 5 and 6 of better quality (more readable).

Response 9: Dear reviewer, figure 1 has been modified to follow the journal's format.

Point 10: Please check the values of added pesticide in Table 3; especially for: Triazophos and Indoxacarb

Response 10: Dear reviewer, sorry for not being serious about our writing, it has been revised.

Point 11: Table 5: Please provide references for the standards in the countries listed.

Response 11: Dear reviewer, It was added in section 3.6.

Point 12: Line 272: Please give literature reference for ADI.

Response 12: Dear reviewer, it has been revised.

Point 13: Table 6: Please provide references for ADI for pesticides (below the table).

Response 13: References have been added to the header of table 6.

Point 14: Did Pu'er tea samples from three different regions show differences in pesticide content?

Which regions of the Pu'er tea samples had a higher pesticide content?

Response 14: Dear reviewer, there were differences in pesticides in Pu'er tea samples from three different regions, the average content of chlorpyrifos: Xishuangbanna > Pu'er > Lincang, the average content of indoxacarb: Xishuangbanna > Lincang > Pu'er, the average content of triazophos: Xishuangbanna > Pu'er > Lincang , the average content of tolfenpyrad: Pu'er > Xishuangbanna > Lincang. In general, the average pesticide content in Xishuangbanna is higher than that in the other two regions.

Reviewer 6 Report

The study was carried out to determine the health risk assessment of only four pesticides (chlorpyrifos, indoxacarb, triazophos and tolfenpyrad) used for the harvest treatment of Pu'er tea from Yunnan, China.

Initially, a QuEChERS method was optimized in terms of the dSPE cleanup step and it was adequately validated combined with UHPLC/QTRAP-MS/MS. Simultaneous determination of these pesticides was then performed in 300 samples. The authors highlighted that the risk was considered acceptable for all four pesticides in the samples collected.

In my opinion, this type of study is not new. In fact, there are many similar studies applied to a wide variety of crops. Moreover, only 4 pesticides were studied. But what is even more important, the use of bare-CNTs in the dSPE step of the QuEChERS method was introduced 10 years ago:

https://doi.org/10.1002/jssc.201100566

Currently, efforts are being made to functionalize carbon nanotubes to increase their selectivity (the authors of this study did not include a current state of the art):

https://doi.org/10.1016/j.foodchem.2020.127805

https://doi.org/10.1021/acs.jafc.9b00090

https://doi.org/10.1016/j.foodcont.2021.108436

https://doi.org/10.1016/j.foodcont.2021.108168

https://doi.org/10.1016/j.jfca.2021.103980

With all the above, I cannot recommend publishing this article in Molecules.

In addition to the above:

- Extensive editing of English language and style is required. There are lots of grammar errors in the manuscript.

- The word QuEChERS should be included as a keyword and it should be defined in the introduction. In fact, the use of acronyms needs revision (GCB and PSA have not been defined in line 102, "limits of quantification has been written after defining the acronym LOQ in line 243, etc.), as well as, the number format of NH2 and C18 should be subscripted.

- The acronym for carbon nanotubes is usually CNTs.

- Lines 24-26: triazophos does not appear.

- The total number of samples analyzed, and the ranges of residue concentrations found should be included in the abstract.

- Line 41 “and excessive use of pesticides has led to Pu'er tea contains a lot of 41 pesticide residues”. The authors should cite the references that support this affirmation. In the same way, the authors should cite references that support the statements included between lines 53 and 63 regarding the disadvantages of these analytical techniques.

- Line 84: Authors should indicate when the samples were collected.

-Line 99: Is the MgSO4 used anhydride? Why didn't they use it in the extraction step (only used it in the cleanup step) as is usual in the QuEchERS method?

- It is recommended to separate the sample preparation method, the instrumental analysis method, and the method validation into three different consecutive sections.

- L- Line 138: The SANTE/11813/2017 guide is not the current version. Document No. SANTE/12682/2019 was implemented by 01.01.2020.

-Lines 146-148: The authors should also include negative values. Matrix effect also occurs for values greater than 50% or less than -50%.

- Line 161: “a T3 chromatographic column with better separation performance than C18 was selected”. The authors should show evidence from the comparative study with these pesticides to support this claim.

-Figure 1: Tea samples are relatively complex, so it would be convenient to include the chromatograms of all four pesticides in the matrix extract.

- Line 204: “In chlorpyrifos, the slopes of acetonitrile extract and MWCNTs purification solution were closer to methanol.” However, in the graph it seems that it is the GCB curve (grey) and not the matrix extract (ochre). In the case of triazophos it seems to be that of C18 and not florisil.

- Line 211. “and MWCNTs can be effective to remove the pigment in tea”. The authors also do not show evidence of pigment reduction such as a photograph. Why did the authors not determine the amount of coextracted material obtained after the cleaning step for each of the sorbents used? This is common practice when it comes to studying a cleaning step since, in addition to the matrix effect, it is important to know the amount of co-extracted material that is injected into the instrument. They could also have done it by studying the amount of MWCNTS, in such a way that graph 5 had two entries simultaneously: recoveries and amount of coextracted removed.

- Line 241: “Linearity was studied in the range 0.01–5.0 ng/mL for four pesticides by matrix-matched standard calibration”. Dado que la matriz es sólida, el rango de linealidad del calibrado en la matriz se debería expresar en µg/kg.

- The format of the references is not according to the standard of the journal.

Author Response

Point 1: In my opinion, this type of study is not new. In fact, there are many similar studies applied to a wide variety of crops. Moreover, only 4 pesticides were studied. But what is even more important, the use of bare-CNTs in the dSPE step of the QuEChERS method was introduced 10 years ago.

Response 1: China's Pu'er tea is one of the famous teas in the world. Yunnan Province is the main producing area of Pu'er tea, and the export volume of Pu'er tea is relatively large every year. The complex matrix of Pu'er tea contains a variety of chemical components, the strong matrix effects affect the detection of pesticide. The qualitative analysis for the low amount of pesticides in Pu'er tea was difficult especially when there is interference from the transitions or a shift of retention times. Therefore, the detection methods of four pesticides with high detection rate such as chlorpyrifos, indoxacarb, triazophos and tolfenpyrad in Pu'er tea were established using QuEChERS, combined with UHPLC/QTRAP-MS/MS and MWCNs. Interfering substances such as pigments, polyphenols and acidic components in Pu'er tea can be effectively removed by MWCNTs, and effectively reduce the matrix effect. The MWCNTs used in this experiment have rarely been reported in the detection of pesticides in Pu'er tea, and combined with QTrap technology can better improve the confirmation of detection.

Point 2: Extensive editing of English language and style is required. There are lots of grammar errors in the manuscript.

Response 2: Dear reviewer, it has been revised.

Point 3: The word QuEChERS should be included as a keyword and it should be defined in the introduction. In fact, the use of acronyms needs revision (GCB and PSA have not been defined in line 102, "limits of quantification has been written after defining the acronym LOQ in line 243, etc.), as well as, the number format of NH2 and C18 should be subscripted.

Response 3: Dear reviewer, sorry for not being serious about our writing, it has been revised.

Point 4: The acronym for carbon nanotubes is usually CNTs.

Response 4: Dear reviewer, multi-walled carbon nanotubes (MWCNs) were used in this study, not carbon nanotubes (CNTs).

Point 5: Lines 24-26: triazophos does not appear.

Response 5: Since China's National food safety standard—Maximum residue limits for pesticides in food (GB 2763-2021) does not stipulate the limit standard of triazophos in tea, it is impossible to judge whether triazophos exceeds the limit standard.

Point 6: The total number of samples analyzed, and the ranges of residue concentrations found should be included in the abstract.

Response 6: Dear reviewer, it has been revised.

Point 7: Line 41 “and excessive use of pesticides has led to Pu'er tea contains a lot of 41 pesticide residues”. The authors should cite the references that support this affirmation. In the same way, the authors should cite references that support the statements included between lines 53 and 63 regarding the disadvantages of these analytical techniques.

Response 7: Dear reviewer, it has been revised.

Point 8: Line 84: Authors should indicate when the samples were collected.

Response 8: Dear reviewer, it has been revised.

Point 9: Line 99: Is the MgSO4 used anhydride? Why didn't they use it in the extraction step (only used it in the cleanup step) as is usual in the QuEchERS method?

Response 9: Dear reviewer, MgSO4 used anhydrous, not anhydride. This experiment was based on the conventional QuEchERS method. According to the Pu'er tea matrix, it is not necessary to add buffer salts during the extraction process. The recovery, precision and stability experiments show that this method meets the relevant detection requirements.

Point 10: It is recommended to separate the sample preparation method, the instrumental analysis method, and the method validation into three different consecutive sections.

Response 10: Dear reviewer, it has been revised.

Point 11: Line 138: The SANTE/11813/2017 guide is not the current version. Document No. SANTE/12682/2019 was implemented by 01.01.2020.

Response 11: Dear reviewer, it has been revised.

Point 12: Lines 146-148: The authors should also include negative values. Matrix effect also occurs for values greater than 50% or less than -50%.

Response 12: Dear reviewer, it has been revised.

Point 13: Line 161: “a T3 chromatographic column with better separation performance than C18 was selected”. The authors should show evidence from the comparative study with these pesticides to support this claim.

Response 13: Dear reviewer, relevant references have been added.

Point 14: Figure 1: Tea samples are relatively complex, so it would be convenient to include the chromatograms of all four pesticides in the matrix extract.

Response 14: Dear reviewer, the graphs in Figure 1 were obtained from standard solutions prepared with methanol. Most of the literatures refer to the use of solvent standards. Therefore, the chromatograms of solvent standards were also used in this paper. If a standard chromatogram of matrix extract solution is required, please let us know in time.

Point 15: Line 204: “In chlorpyrifos, the slopes of acetonitrile extract and MWCNTs purification solution were closer to methanol.” However, in the graph it seems that it is the GCB curve (grey) and not the matrix extract (ochre). In the case of triazophos it seems to be that of C18 and not florisil.

Response 15: Dear reviewer, sorry for misjudging GCB curve as the matrix extract and C18 curve as florisil curve, it has been revised.

Point 16: Line 211. “and MWCNTs can be effective to remove the pigment in tea”. The authors also do not show evidence of pigment reduction such as a photograph. Why did the authors not determine the amount of coextracted material obtained after the cleaning step for each of the sorbents used? This is common practice when it comes to studying a cleaning step since, in addition to the matrix effect, it is important to know the amount of co-extracted material that is injected into the instrument. They could also have done it by studying the amount of MWCNTS, in such a way that graph 5 had two entries simultaneously: recoveries and amount of coextracted removed.

Response 16: Dear reviewer, the effect of different adsorbents on pigment removal was shown below. The adsorbents used from left to right in the figure 1 were Florisil, GCB, NH2, C18, PSA and MWCNTs. It can be seen from the dark and light colors that Florisil, C18 and PSA has poor pigment removal effect, GCB and MWCNTs have better pigment removal effect, combined with the results of matrix effect, MWCNTs was finally selected as the best adsorbent.

Figure 1. The effect of different adsorbents on pigment removal (from left to right in the figure were Florisil, GCB, NH2, C18, PSA and MWCNTs)

Sorry for the incompleteness of our experimental design. In this experiment, only the matrix effect was studied, and the amount of coextracted removed was not considered. This may be something that needs to be considered in future research. Thanks to the reviewers for their good suggestions.

Point 17: Line 241: “Linearity was studied in the range 0.01–5.0 ng/mL for four pesticides by matrix-matched standard calibration”. Dado que la matriz es sólida, el rango de linealidad del calibrado en la matriz se debería expresar en µg/kg.

Response 17: Dear reviewer, the preparation method of matrix-matched standard calibration in this experiment was to use the matrix extract to dilute the standard solution. Different from the solvent standard calibration, the matrix-matched standard calibration solution is prepared using the matrix extract. Therefore, it is not solid, nor should it be μg/kg.

Point 18: The format of the references is not according to the standard of the journal.

Response 18: Dear reviewer, the format of the references was revised.

Round 2

Reviewer 2 Report

The authors fulfilled all the requests and improved the manuscript. Therefore, in my opinion it is now suitable for publication

Reviewer 4 Report

 The paper “Simultaneous determination and health risk assessment of four high detection rate pesticide residues in Pu'er tea from Yunnan, China” is good and all the requests and suggestions were completed exhaustively. There is only one last suggestion regarding the figure 3 and 4. The resolution is still too low because it is not possible to read the numbers on top of the bars for the figure 4 and the labels inside the graphs of figure 3.

Reviewer 5 Report

The authors significantly improved the manuscript which can be accepted now.

Reviewer 6 Report

Despite the lack of novelty, the manuscript has improved considerably and can be accepted.

This manuscript is a resubmission of an earlier submission. The following is a list of the peer review reports and author responses from that submission.